# Targeted Therapy in HR+ HER2− Metastatic Breast Cancer: Current Clinical Trials and Their Implications for CDK4/6 Inhibitor Therapy and beyond Treatment Options

**DOI:** 10.3390/cancers13235994

**Published:** 2021-11-29

**Authors:** Constanze Elfgen, Vesna Bjelic-Radisic

**Affiliations:** 1Breast Surgery, Breast-Center Zurich, 8008 Zurich, Switzerland; 2Faculty of Medicine, University of Witten-Herdecke, 58455 Witten, Germany; vesna.bjelic-radisic@helios-gesundheit.de; 3Institute of Gynecology and Obstetrics, University Hospital Wuppertal, 42109 Wuppertal, Germany

**Keywords:** CDK4/6 inhibitor, PI3K inhibitor, quality of life in MBC, metastatic breast cancer, targeted therapy, HR-positive breast cancer

## Abstract

**Simple Summary:**

In the treatment of hormone-receptor positive, HER2 negative metastatic breast cancer, targeted therapy showed improved overall survival and it has become an established treatment within recent years. Some study results conflict with others. As multiple new research articles on this topic have been recently published, this review aims to crystallize the current relevant results.

**Abstract:**

A metastatic state of breast cancer (MBC) affects hundreds of thousands of women worldwide. In hormone receptor-positive (HR+)/human epidermal growth factor receptor 2-negative (HER2−) MBC, cyclin-dependent kinase (CDK)4/6 inhibitors can improve the progression-free survival (PFS), as well as the overall survival (OS), in selected patients and have been established as first- and second-line therapies. However, as MBC remains uncurable, resistance to CDK4/6 inhibitors occurs and requires alternative treatment approaches. Data on targeted therapy continue to mature, and the number of publications has been constantly rising. This review provides a summary and update on the clinical relevance, patient selection, ongoing trials of CDK4/6 inhibitors, and further targeted therapy options. It focuses on clinical aspects and practicability, as well as adverse events and patient-reported outcomes.

## 1. Introduction

Breast cancer (BC) is the most frequent malignant disease in women worldwide, and more than one in five affected women experience a metastatic stage, which is still incurable [1]. Around 75% of BC patients are diagnosed with a hormone receptor-positive (HR+), human epidermal growth factor receptor 2-negative (HER2−) type of BC [2]. For these patients, endocrine therapy is a major approach for systemic treatment. Aromatase inhibitors (AI), selective estrogen receptor degraders (SERDs), and selective estrogen receptor modulators (SERMs) play an essential role in this context [3]. However, in metastatic or advanced breast cancer (MBC/ABC), resistance to endocrine therapy occurs, which partly explains the poor median overall survival (OS) of less than five years after metastasis has been diagnosed [4]. Cancer cells underly a high rate of genetic mutations and dysregulations of the cell-dividing cycle, and the development of drugs that interfere in these dysregulations without affecting the whole organism has been an ambitious project. This is especially true for patients with HR+ MBC. Cyclin-dependent kinase 4/6 (CDK4/6) inhibitors and phosphoinositide 3-kinase (PI3K) inhibitors act as targeted therapies in HR+ MBC and have shown promising results in clinical studies [5,6]. However, some expectations for these drugs could not be fulfilled. As the number of new peer-reviewed publications concerning CDK4/6 inhibitors for the therapy of MBC has risen in recent years, limitations have arisen in real-life settings. Furthermore, mature OS data from earlier significant trials have recently been published. Our review aims to crystallize the current role of CDK4/6 inhibitors in MBC, considering clinical studies, real-world data, toxicity, and quality of life (QOL). Furthermore we discuss alternative therapeutic approaches in the case of CDK4/6 inhibitor resistance. The review may help oncologist practitioners to determine which patients will benefit from the therapy and those which will probably not.

## 2. The Rationale of Targeting CDK4/6 for Breast Cancer Therapy

Briefly, CDKs are enzymes that play a central role in cell cycle progression. The activated complex of D-type cyclins and CDK4/6 leads to phosphorylation and therefore inactivation of retinoblastoma-associated protein. This process is regulated by genes that are a prerequisite for S-phase entry and cell division [7]. These mechanisms are known in all types of BC. However, estrogen acts as a stimulator of this cascade as activated ER increases the amplification of D cyclins, mitogenic enzymes that are associated with multiple cancers [8]. As the expression of cyclin D is high in HR+ breast cancer, this G1-to-S checkpoint represents an ideal therapeutic target. D-type cyclins bind to CDK4/6, and a deregulation of the CDK4/6 pathway can often be observed in HR+ BC (Figure 1) [7,8,9]. The rationale of pharmaceutical intervention is to inhibit CDK4/6 in HR+ BC, and therefore to interrupt the activating mechanisms and trigger cell cycle arrest. As a first- or second-line therapy in HR+, HER2− MBC, three small molecules inhibitors of CDK4/6 with similar mechanisms of action have been approved by the FDA: palbociclib, ribociclib, and abemaciclib [10].

The use of CDK4/6 inhibitors is standard in combination with the endocrine therapy of an AI, whereas a combination with fulvestrant is preferable in patients who show progressive disease or relapse under AI [11]. Abemaciclib has also been approved as an monotherapeutic agent in heavily pretreated HR+, HER2− MBC patients [12]. CDK4/6 inhibitors have significantly improved the progression-free survival (PFS) by several months, compared to endocrine treatment alone, in prospective, randomized clinical trials (Table 1) [13]. Differentiating an OS benefit is challenging in an unselected group of patients with MBC because of the heterogeneity of patients and clinical metastatic manifestations, as well as the variability of previous and subsequent therapies. Furthermore, dose modification and the discontinuation rate may vary much more in patients with MBC than in patients with early BC [14,15]. All of these aspects complicate the meta-analysis of OS in pivotal clinical trials. Multiple earlier studies and their interim analyses were unable to show a significant improvement in OS when adding a CDK4/6 inhibitor to endocrine therapy (Table 1). As mature OS data containing many years subsequent to the beginning of randomization have been published, we now have a clearer picture of CDK4/6 inhibitors. Recently, the MONALEESA-3 trial revealed a significantly improved OS in MBC patients who received ribociclib plus fulvestrant versus placebo plus fulvestrant (33.6 versus 19.2 months, respectively) [6]. Likewise, the MONALEESA-7 trial emphasized an OS benefit in selected pre-/perimenopausal women who were not heavily pre-treated (as only 14% received chemotherapy in the advanced setting) [16]. Follow-up data from the MONARCH-2 trial also showed a longer OS in the abemacilib arm (46.7 versus 37.3 months) [17]. However, as the benefit in OS data has been contradictory in previous clinical trials, the question arises as to how patients with an expected benefit can be selected. No significant OS benefit of the third approved CDK4/6 inhibitor (Palbociclib) has been shown to date [18].

The data are unclear for the subtypes of MBC other than HR+, HER2−. In HER2+ BC, the oncogenetic activation of the membrane thyrosine kinase HER2 promotes pathogenesis and the progression of cancer cells by interacting with the PI3K/AKT/mTOR pathway and other signaling pathways [19]. Targeted anti-HER2 therapies such as trastuzumab remarkably improved the outcome of HER2+ BC patients. However, resistance to these therapies has been observed and can be mediated by the cyclin D1-CDK4 pathway; inhibiting this axis might decrease the resistance [19]. Despite promising results from earlier transgenic mouse models, the clinical relevance of CDK4/6 inhibitors in HER2+ MBC still has to be defined [19]. Recent data from the monarcHER trial showed an improved PFS in HR+, HER2+ MBC patients who received abemaciclib plus fulvestrant plus trastuzumab versus standard-of-care chemotherapy plus trastuzumab [20]. Despite these encouraging results, it has to be qualified with the fact that a control group with endocrine therapy plus trastuzumab was lacking. As multiple trials have been planned and are already running, it is thought that clearer answers to the question of clinical relevance will be provided in the near future (Table 2). This is also true for patients with triple-negative breast cancer (TNBC) (Table 2). Given the mechanism of action of CDK4/6 inhibitors, a therapeutic effect in the heterogeneous group of TNBC is not expected. Nevertheless, in vitro and clinical studies suggest a potential benefit of CDK4/6 inhibitors as a pre-treatment to chemotherapy for subgroups of TNBC, especially in cell lines showing the expression and activation of cyclin D1, CDK4/6, and Rb-proficiency [21].

## 3. Clinical Impact and Real-Life Data on CDK4/6 Inhibitors

Beyond prospective, randomized, double-blind studies, multiple clinical real-life studies considering CDK4/6 inhibitors in MBC have been published within recent years. In real-world studies, nonrandomized heterogeneous groups of patients with different pre-treatments and co-morbidities who present in the clinical routine are included and therefore the selection bias is lower [22]. Furthermore, data collection for patient subgroups who are excluded or underrepresented in randomized, controlled clinical trials is achievable. Many topics related to practical aspects of the CDK4/6 inhibitor therapy are still unclear, and with the rising numbers of studies, optimism exists that clear answers will be found in the future. Firstly, we do not know at what point of disease progression a CDK4/6 inhibitor therapy should be started. Considering an early relapse (<12 months) as a sign of resistance to endocrine therapy, an addition of a CDK4/6 inhibitor is indicated in these patients [13,23]. However, an improved PFS was also observed in clinical trials with first-line CDK4/6 inhibitor therapy [24]. Despite the fact that patients having a better prognosis, especially with bone-only metastasis, usually respond well to endocrine-only therapy, there is potential for the improvement of PFS with a CDK4/6 inhibitor. Ongoing and future clinical studies will help in finding an answer to the first-/second-line question in relation to CDK4/6 inhibitors [25] (Table 2).

A recent meta-analysis showed no superiority of a chemotherapy regimen, with or without targeted agents, compared to CDK4/6-inhibitor-plus-endocrine therapy in postmenopausal patients with HR+, HER2− MBC [26]. Because of the favorable toxicity profile, CDK4/6 inhibitors should be preferred over chemotherapy in first- or second-line therapy; except in the presence of visceral crisis.

Current pooled data analyses emphasize a benefit in PFS for all analyzed subgroups of HR+, HER2− MBC patients [27,28]. The benefit occurred regardless of the combined endocrine drug (AI/fulvestrant), line of therapy, site of metastases, presence of visceral metastases, or the length of the treatment-free interval. Furthermore, there is evidence to suggest an improvement of PFS independent of age and menopausal status. However, data suggest that some subgroups of patients have a higher benefit than others, mainly in the subgroups of postmenopausal women, patients with visceral metastasis, and patients with the progression of disease under endocrine therapy (second-line therapy) [29]. Regarding the intrinsic molecular subtypes of BC, PFS is marginally pronounced in the luminal A subtype compared to the luminal B subtype in the cohort of PALOMA-2 and -3 [30].

Due to the immaturity of the OS data in most of the considered studies, the interpretation of OS improvement needs to be regarded with caution [27]. To date, we have seen statistically relevant longer OS in the few studies with abemaciclib and ribociclib, but not with palbocilicib, which only showed a positive trend [28]. Despite meta-analyses suggesting a PFS benefit for all HR+, HER2− MBC patients, it is worth considering the subgroup analyses from separate pivotal studies. As already mentioned, the group of patients with HR+, HER2− MBC is heterogeneous, and if a subgroup analysis reveals a superior benefit, the question arises whether this subgroup has additional selection bias.

Regarding menopausal status, premenopausal women are underrepresented in most clinical studies of MBC. The MONALEESA-7 study selected premenopausal women with HR+, HER2− MBC and revealed a significant improvement of PFS and OS in the ribocilib group (OS at 42 months of 70.2% versus 46.0% in the placebo group) [16]. PFS was shorter in patients with prior chemotherapy. No strong benefit could be observed in women with bone-only metastasis or progression > 12 months after the end of the endocrine therapy, as these patients have a significant better prognosis per se [16]. Similarly, in the MONARCH-2 trial, including pre- and postmenopausal women, patients with visceral metastases and patients with primary endocrine therapy resistance, both poor prognostic factors, had a stronger OS effect [17]. Interestingly, an analysis from the PALOMA-3 trial showed an improved OS in the subgroup of patients with endocrine-sensitive tumors, as well as in the subgroup without prior chemotherapy in the MBC status [31]. Patients with metastasis of the central nervous system may benefit from abemaciclib, as it penetrates the blood–brain barrier. As patients with CNS metastasis were excluded from most phase III CDK4/6 inhibitor trials, a clear recommendation cannot be given. However, a current study revealed potential benefit for patients with HR+, HER2− MBC with brain metastasis who were naïve to a CDK4/6 inhibitor therapy [32].

We still know far too little about predictive biomarkers for CDK4/6 inhibitors, and therefore the data and analyses only partially explain the differing benefits of MBC patient subgroups. As the relevance of individualized medicine increases, studies considering the complexity of patient-related and tumor-related factors are crucially needed [33,34] (Table 2).

## 4. Adverse Events, Quality of Life, and Non-Compliance under CDK4/6 Inhibitors

Prolongated PFS and OS under palliative therapy always need to be balanced between toxicity and the risk of adverse events (AEs) on the one hand, and the potential benefit of less morbidity and gained lifetime on the other hand. The tolerability of CDK4/6 inhibitors is usually acceptable and manageable with dose modification and side effect treatment. Minor AEs are reported in most patients (60–80%) under CDK4/6 inhibitor therapy, resulting in a dose reduction in every third patient [35,36]. Serious AEs (SAEs; grade 3/4) that lead to discontinuation are reported in up to 12% of patients [36]. Hematological toxicities are frequent, especially in palbociclib and ribociclib, leading to neutropenia, anemia, and thrombocytopenia [37]. Compared to most chemotherapy regimens, toxicity is lower, and AEs are manageable in most cases with early interventions. Fatigue, diarrhea, and nausea are more often reported under abemaciclib intake, leading to a significant higher rate of treatment discontinuation [38]. A pronunciation of gastrointestinal AEs is observed in combination with fulvestrant compared to AI. Less frequent AEs are hepatotoxicity, venous thromboembolic events, QT-interval prolongation, increased serum creatinine, and pneumonitis [38]. The high risk of SAE should be especially considered in long-term treatment. As all three CDK4/6 inhibitors showed a comparable improvement of PFS, treatment choices can consider the drug with the most reasonable side effect profile for the individual patient [26]. This can be recommended as long as mature OS data for the single drugs are not present. In case of the proven superiority of one or two CDK4/6 inhibitors, this up-front recommendation will change.

Fortunately, the measurement of patient-reported outcome (PRO) has become an important tool in prospective, randomized trials. However, quality-of-life (QOL) data in CDK4/6 inhibitors are still limited. The available data suggest a satisfactory-to-good QOL in patients with MBC and CDK4/6 inhibitor therapy, even when experiencing AEs [35,39]. In the setting of the PALOMA-2 trial, decreasing QOL was mainly associated with progressive disease and a non-response [40]. Beyond clinical trials, we observe noncompliance with palliative systemic therapy in a notable group of unselected patients with MBC [41]. However, there has been no meta-analysis of compliance in CDK4/6 inhibitor therapy, and real-life clinical studies report disruption of the therapy in 11% and non-adherence in more than 20% of cases, without considering patients with initial therapy refusal [42]. The reasons for non-compliance and non-adherence are heterogeneous and closely based on the individual patient’s experience and background. Objectively “light” side effects might be intolerable in some patients, whereas other patients are highly motivated to continue therapy even under SAE. In understanding the patient’s individual needs, support and compliance can be improved.

## 5. Resistance to CDK4/6 Inhibitors

CDK4/6 inhibitor therapy can prolong the time of stable disease in HR+ MBC. However, primary resistance to CDK4/6 inhibitors occurs in about 15–30% of cases, and at some point of the therapy, almost all patients develop a progressive disease, which reflects an acquired resistance to CDK4/6 inhibitors [43]. Interestingly, a therapeutic switch to abemaciclib in patients who progress under other CDK4/6 inhibitors was beneficial for a subgroup of patients with HR+, HER2− MBC [44]. As is the case in other clinical studies, potential predictive biomarkers are missing. Preclinical and clinical studies suggest variable mechanisms of tumor cell resistance due to several tumor suppressor gene mutations, but without evident predictive value [45]. PIK3CA and ESR1 mutations can be detected in circulating tumor DNA in up to every third patient with HR+, HER2− MBC [46]. However, several studies, such as the PALOMA-3 trial, did not reveal a predictive value of the CDK4/6 inhibitor response [43]. Whole-exome sequencing of 59 tumors suggests an association of several mutations (loss of RB1, loss of estrogen receptor, activating alterations in AKT1, RAS, AURKA, CCNE2, ERBB2, and FGFR2) to CDK4/6 inhibitor resistance, but still without predictive value [47]. Interestingly, the prevalence of BC cells with estrogen receptor 1 (ESR1) mutations is much higher in patients with MBC who received AI (up to 40% versus 1% in MBC patients without prior ET). Despite a clear association between ESR1 mutations and endocrine resistance, data about predicting the response to AI and/or CDK4/6 inhibitors are contradictory [48]. Further studies are warranted to clarify whether the detection of ESR1 mutations may influence clinical decisions and the indication of CDK4/6 inhibitors in the metastatic or adjuvant setting [49]. Recently, tumor mutations of the KRAS protein that induce cyclin D1 overexpression have been associated with acquired loss of responsiveness [50]. As in the cited study where circulating tumor cells in the plasma were analyzed, liquid biopsy could be a potential clinical approach to predict resistance to CDK4/6 inhibitors. It is assumed that the complex mechanism of acquired resistance is not yet fully understood. The only predictive biomarker for CDK4/6 inhibitors remains a positive estrogen receptor status; further biomarkers that predict sensitivity for or resistance to CDK4/6 inhibitors are still missing [27,34,51].

## 6. Alternative Therapeutic Approaches for Patients with CDK4/6 Inhibitor Resistance

On a molecular level, the CDK4/6 and PI3K-mTOR pathways interfere with the cell signaling and division processes. Both pathways are closely related to each other, as a hyperactivated PI3K-mTOR pathway increases cyclin D levels [52]. Physiologically, these pathways regulate cell metabolism, survival, proliferation, and growth in multicellular organisms [53]. Deregulation of the PI3K/AKT/mTOR pathway caused by PIK3CA-activating mutations can be detected in almost every third patient with BC, and in about 40% of patients with HR+, HER2− BC [54,55]. When detected in early HR+ BC, these mutations are associated with improved disease-free survival. In patients with advanced and metastatic HR+ BC, increasing resistance to endocrine therapy and a lower response to chemotherapy was observed when PIK3CA-activating mutations were present [56,57,58].

### 6.1. PI3K Inhibitors

The earlier BELLE trial observed a therapeutic benefit from buparlisib, a PI3K inhibitor without isoform specificity, but with high off-target side effects and without clinical relevance [59]. More specific PI3K inhibitors promise to reduce off-target toxicities; however, taselisib, a PI3Kß-specific inhibitor, still showed an insufficient safety profile and only a modest benefit [60]. The first approved PIK3 inhibitor, alpelisib, selectively inhibits the alpha isoform of PI3K that is encoded by PIK3CA mutations. In the phase III trial SOLAR-1, alpelisib improved the PFS in the cohort with PIK3CA-mutated cancer (11.7 months in the alpelisib-fulvestrant group versus 5.7 months in the placebo-fulvestrant group) [61]. This was especially true for the small group of patients who received previous therapy with a CDK4/6 inhibitor plus AI [61]. These findings are emphasized by recently published interim results from the BYLieve trial with a PFS of 50.4% after six months [5]. The observed cohort received alpelisib after disease progression on or after a CDK4/6 inhibitor plus AI. However, the number of patients was limited, and a comparator group was lacking. Discontinuation due to AE was reported in every fifth patient. Overall, the safety profile in the BYLieve trial was tolerable under careful AE monitoring, especially of hyperglycemia. Alpelisib is a specific PI3K inhibitor that targets the catalytic alpha-unit of PI3K, but as these drugs are ATP-competitive, they also affect a major signaling cell pathway [62]. This mechanism of action explains concentration-dependent and frequent severe AEs. Skin rash, diarrhea, alopecia, nausea, and hyperglycemia are the most common side effects of alpelisib [61]. Threatening hyperglycemia and ketoacidosis have been reported in less than 1% of cases but this remains a worrisome AE that requires close monitoring during drug administration [63]. Obviously, alpelisib should be restrictively indicated in patients with diabetes. The patient-reported outcome in the SOLAR-1 study showed no statistical difference between arms. Despite this, it is an important consideration that patients participating a phase III trial are carefully selected and monitored; these results support the manageable risk profile of alpelisib [64]. Only very limited real-world data on alpelisib in the clinical routine are available so far. Recently published results emphasized the importance of careful interdisciplinary surveillance and patient education under alpelisib therapy [65].

Stable disease was significantly improved in patients with HR+, PIK3CA-mutated ABC who previously experienced progression on or after CDK4/6 inhibitor plus AI [5]. CDK4/6 inhibitors sensitize cells with resistance to PI3K inhibitor in vivo [52]. On the other hand, a clinical study suggests cross-resistance to PI3K inhibitors of tumors previously treated with CDK4/6 inhibitors, which is probably based on the loss of PTEN [45]. In patients with a PIK3CA-mutated HER2+ BC, the therapy combination with trastuzumab was limited by severe gastrointestinal toxicity [66]. Likewise, a combination with chemotherapy in patients with PIK3CA-mutated TNBC has no clinical relevance due to the toxicity profile [67]. As already emphasized, PIK3CA mutations cannot act as a biomarker of response or resistance to CDK4/6 inhibitors. However, there is evidence to suggest that PIK3CA mutations are acquired during endocrine treatment alone, as well as during CDK4/6 inhibitor plus endocrine therapy [68]. Interestingly, an association between reduced CDK4/6 inhibitor sensitivity and PI3K mutations detected in liquid biopsy was observed in a pilot study [69]. The SOLAR-1 trial, as well as the SANDPIPER trial, showed no relevant benefit in patients without PIK3CA-mutated MBC [61,70]. In HR+, HER2− MBC patients with resistance to CDK4/6 inhibitors plus endocrine treatment, PIK3CA mutation testing on tumor tissue or circulating tumor cells is justified to select patients who could potentially benefit from alpelisib [71]. To date, no predictive markers have been found that indicate a clinical response to PI3K inhibitors in the subgroup of patients with PIK3CA-activating mutations [52]. Multiple ongoing and future clinical trials are expected to clarify the relevance of alpelisib and potentially other PI3K inhibitors in the clinical routine (Table 3).

### 6.2. Everolimus

Endocrine resistance is more frequent in tumors with a hyperactivated PAM pathway, which can be caused by mutations of the PI3K gene and AKT activation [72]. The mammalian target of the rapamycin (mTOR) inhibitor everolimus was the first targeted drug in patients with endocrine resistance and showed a prolonged PFS in patients with HR+, HER2− MBC. Moreover, it has a positive effect on bone metabolism and delays bone metastasis progression [73]. An important issue in everolimus therapy is the toxicity profile, which is less favorable compared to that of CDK4/6 inhibitors. The most frequent AEs are hyperglycemia, stomatitis, anemia, dyspnea, and fatigue and this led to dose reduction in two-thirds of patients and a therapy discontinuation in more than 12% in several studies [74]. However, a network analysis showed a similar significant improvement of PFS in studies with CDK4/6 inhibitor- based combinations compared with studies with everolimus plus exemestane; no randomized trial of both these combinations has been performed [75]. An improvement of OS was observed in randomized phase III studies with everolimus, but without statistical relevance [76]. In contrast, a clearly relevant benefit was demonstrated in recent CDK4/6 inhibitor trials [6,16]. It is notable that in the BOLERO-2 study, the benefit from everolimus was not influenced by the presence of PIK3CA tumor mutations [77]. A recent retrospective comparative analysis showed a higher efficacy of everolimus plus exemestane in patients with prior CDK4/6 inhibitor plus endocrine therapy than in patients with prior endocrine therapy alone [78]. A potential selection bias caused by patients with CDK4/6 inhibitor resistance due to PIK3CA mutations who benefit from a second- or third-line therapy with everolimus has to be considered. However, everolimus remains a second- or third-line treatment option in patients with HR+, HER2− MBC. This is especially true for patients with PIK3CA mutations who show a disease progression on PI3K inhibitor therapy or who experience contraindications to PI3K inhibitors [78].

### 6.3. AKT Inhibitors

A further option for patients with resistance to CDK4/6 inhibitors could be treatment with AKT inhibitors, as recently shown by results from the FAKTION trial [79]. Capivasertib, a selective inhibitor of the serine/threonine kinase AKT, plus fulvestrant showed a significant improvement of PFS compared to placebo plus fulvestrant in patients with aromatase inhibitor- resistant MBC. However, long-term outcomes and phase III study results with a larger number of patients are still pending.

## 7. Conclusions and Perspective

With the goal of overcoming endocrine resistance in HR+ MBC, the connecting CDK and PI3K/AKT/mTOR pathways were revealed as key points for targeted therapies. As multiple studies have shown a clear benefit of CDK4/6 inhibitors, current guidelines recommend this therapy in patients with MBC [80,81]. The toxicity profile of CDK4/6 inhibitors is favorable to chemotherapy. In patients with primary or acquired resistance to CDK4/6 inhibitors, the earlier-approved everolimus remains an alternative. In the presence of PIK3CA-mutated BC cells, PI3K inhibitors can be indicated. However, data showing a benefit in overall survival are still missing.

The role of targeted therapy in the adjuvant setting is still unclear and is a topic of ongoing studies (Table 4). The PALLAS study did not show a benefit of adding palbociclib to adjuvant endocrine therapy [82]. Similarly, beneficial results were missing in patients with residual disease of HR+, HER2− BC after NACT (PENELOPE-B trial) [83], whereas in patients with high-risk BC in the adjuvant setting, PFS was improved when adding abemaciclib to endocrine therapy [84]. Likewise, in high-risk BC patients with PIK3CA-mutated tumors, there is a benefit associated with PI3K inhibitors in a neoadjuvant approach. The LORELEI study demonstrated a significantly higher response rate in patients treated with teselisib and letrozole versus placebo and letrozole [85].

Our review provides updates on the feasible current targeted therapies in the clinical practice of HR+, HER2− MBC. In summary, adequate patient selection is crucial for therapeutic efficacy. We believe that remaining questions will be answered within the coming years, when selection criteria on a clinical and molecular basis are fully understood.

## 8. Strengths and Limitations

Our review has some limitations. We exclusively considered publications in English. As we included results from real-life studies as well as data from observational studies with patient-reported outcomes, our review is not appropriate for a meta-analysis. However, besides the up-to-date summery of OS and PFS, our review also considers clinically relevant studies about toxicity and quality of life in targeted therapy in patients with HR+ MBC. It discusses the indications, expectations, and limitations of targeted therapy and may help the oncologist to select patients under real-life conditions.

## Figures and Tables

**Figure 1 cancers-13-05994-f001:**
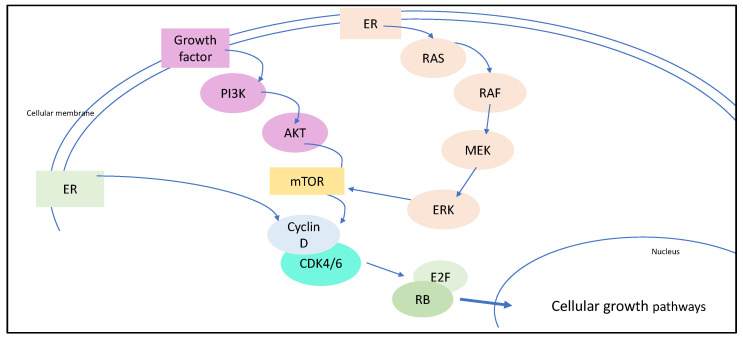
The RAS and PI3K pathway increase mTOR activity which enhance D-type cyclins. The activated complex of D-type cyclins and CK4/6 leads to inactivation of retinoblastoma-associated protein (RB) and therefore to S-phase progression.

**Table 1 cancers-13-05994-t001:** Relevant prospective, randomized clinical trials on CDK4/6 inhibitors in patients with HR+, HER2− metastatic breast cancer.

Clinical Trial	Patient Selection	*n*	Therapeutic Regimen	PFS (Months)	OS
PALOMA-1	Postmenopausal women without systemic treatment for advanced disease	165	Palbociclib + letrozole versus letrozole alone	20.2 versus 10.2 (HR 0.488, 95% CI 0.319–0.748; one-sided *p* = 0.0004)	Not significant
PALOMA-2	Postmenopausal women without systemic treatment for advanced disease	666	Palbociclib + letrozole versus placebo + letrozole	24.8 versus 14.5 (HR 0.58; 95% CI 0.46–0.72; *p* < 0.001)	Immature data
PALOMA-3	Postmenopausal women with progress under endocrine therapy	521	Palbociclib + fulvestrant versus placebo + fulvestrant	9.5 versus 4.6 (HR 0.46; 95% CI 0.36–0.59; *p* < 0.0001)	Not significant
MONARCH-2	Pre- and postmenopausal women with progress under endocrine therapy	669	Abemaciclib + fulvestrant versus placebo + fulvestrant	16.4 versus 9.3 (HR 0.553; 95% CI 0.449 to 0.681; *p* <0.001)	Not significant
MONARCH-3	Postmenopausal women without systemic treatment for advanced disease	493	Abemaciclib + endocrine therapy versus placebo + endocrine therapy	28.2 versus 14.8 (HR 95%; CI 0.418–0.698; *p* < 0.0001)	Immature data
MONALEESA-2	Postmenopausal women without systemic treatment for advanced disease	668	Ribociclib + fulvestrant versus placebo + fulvestrant	25.3 versus 16.0 (HR 0.568; 95% CI 0.457–0.704; *p* < 0.0001)	Immature data
MONALEESA-3	Postmenopausal women without systemic treatment for advanced disease	726	Ribociclib + letrozole versus placebo + letrozole	20.5 versus 12.8 (HR 0.59; 95% CI 0.48–0.73; *p*> 0.001)	57.8% vs 45.9% at 42 months (HR 0.72; 95% CI 0.57–0.92; *p* = 0.00455)
MONALEESA-7	Peri-/premenopausal	672	Ribociclib + endocrine therapy versus placebo + endocrine therapy	23.8 versus 13.0 (HR 0.55; 95% CI 0.44–0.69; *p* > 0.001)	70.2% vs 46.0% at 42 months (HR 0.71; 95% CI 0.54–0.95; *p* = 0.00973)

PFS = progression-free survival; HR = hazard ratio; OS = overall survival; CI = confidence interval.

**Table 2 cancers-13-05994-t002:** Selected ongoing phase II and III trials with CDK4/6 inhibitor therapy.

Clinical Trial	Patient Selection	Estimated Enrollment (*n*)	Therapeutic Regimen	Study Start Year	Estimated Completion Year	Clinical Trial Information
PRESERVE 2	Pre- and postmenopausal women and men with metastatic TNBC	250	Trilaciclib + gemcitabine + carboplatin versus placebo + gemcitabine + carboplatin	2021	2024	NCT04799249
Ribociclib with trastuzumab plus letrozole in postmenopausal HR+, HER2+ ABC Patients	Postmenopausal women with advanced HR+, HER2+ BC	95	Letrozole + trastuzumab + ribociclib	2019	2021	NCT03913234
PATRICIA II	Pre- and postmenopausal women with metastatic HR+ or HR-, HER2+ BC	232	Trastuzumab + Palbociclib + letrozole (only in HR+ patients)	2015	2020	NCT02448420
TOUCH	Pre- and postmenopausal women with early HR+, HER2+ BC	144	Paclitaxel + trastuzumab + pertuzumab versus Palbociclib + letrozole + trastuzumab + pertuzumab	2019	2021	NCT03644186
DETECT V	Pre- and postmenopausal women with metastatic HR+, HER2+ BC	270	Ribociclib + trastuzumab + pertuzumab + endocrine therapy versus trastuzumab + pertuzumab + chemotherapy tailored by ribociclib	2015	2021	NCT02344472
SONIA	Pre- and postmenopausal women with HR+, HER2− advanced/metastatic breast cancer.	1050	Aromatase inhibitor + a CDK4/6 inhibitor 1st line versus Fulvestrant + CDK4/6 inhibitor 2nd line	2017	2022	NCT03425838
AMICA	Pre- and postmenopausal women with HR+, HER2− advanced/metastatic breast cancer and disease control after 1st line chemotherapy	150	Ribociclib + endocrine therapy versus endocrine therapy alone	2018	2022	NCT03555877
MAINTAIN	Pre- and postmenopausal women and men with HR+, HER2− advanced/metastatic breast cancer after progression on anti-estrogen therapy plus CDK4/6 inhibitor	132	Ribociclib + fulvestrant versus placebo + fulvestrant	2016	2021	NCT02632045
ABEMACARE	Pre- and postmenopausal women with HR+, HER2− advanced/metastatic breast cancer and symptomatic visceral metastases or high tumor burden	120	Abemaciclib + endocrine therapy	2020	2024	NCT04681768
PALATINE	Pre- and postmenopausal women with HR+, HER2− advanced/metastatic breast cancer.	200	Palbociclib + endocrine therapy upfront	2019	2023	NCT03870919
KENDO	Pre- and postmenopausal women and men with HR+, HER2− advanced/metastatic breast cancer.	150	A CDK4/6 inhibitor + endocrine therapy versus chemotherapy + endocrine therapy	2017	2022	NCT03227328
FATIMA	Premenopausal women with HR+, HER2− advanced/metastatic breast cancer.	160	Palbociclib + exemestane + goserelin versus exemestane + goserelin alone	2019	2021	NCT02917005
PACE	Pre- and postmenopausal women and men with HR+, HER2− advanced/metastatic breast cancer after CDK and endocrine therapy	220	Palbociclib + fulvestrant + avelumab versus Palbociclib + fulvestrant versus fulvestrant alone	2017	2021	NCT03147287

**Table 3 cancers-13-05994-t003:** Selected ongoing trials with PI3K inhibitors in patients with breast cancer.

Clinical Trial	Patient Selection	Estimated Enrollment (*n*)	Therapeutic Regimen	Study Start Year	Estimated Completion Year	Clinical Trial Information
Alpelisib with endocrine therapy following progression on endocrine therapy	Pre- and postmenopausal women and men with PIK3CA mutant HR+, HER2− MBC	44	Alpelisib + aromatase inhibitor or fulvestrant	2021	2024	NCT04762979
Alpelisib with trastuzumab and pertuzumab as maintenance therapy	Pre- and postmenopausal women and men with HER2+, PIK3CA mutant ABC	588	Alpelisib + trastuzumab + pertuzumab versus placebo + trastuzumab + pertuzumab	2020	2025	NCT04208178
Inavolisib in patients with PIK3CA-mutant, HR+, HER2− locally advanced or metastatic breast cancer	Pre- and postmenopausal women and men with HR+, HER2− PIK3CA mutant A/MBC	400	Inavolisib + palbociclib + fulvestrant versus placebo + palbociclib + fulvestrant	2020	2025	NCT04191499
Alpelisib + nab-paclitaxel in subjects with advanced TNBC who carry either a PIK3CA mutation or have PTEN loss (EPIK-B3)	Pre- and postmenopausal women and men with advanced TNBC with PIK3CA mutation or PTEN loss	566	Alpelisib + nab-paclitaxel versus placebo + nab-paclitaxel	2020	2023	NCT04251533
PERSEVERE	Pre- and postmenopausal women and men with post-neoadjuvant residual TNBC	197	Assignment due to the genomic target of ctDNA: talazoparib + capecitabine; atezolizumab + capecitabine; inavolisib + capecitabine; talazoparib + atezolizumab + capecitabine	2021	2024	NCT04849364

**Table 4 cancers-13-05994-t004:** Selected ongoing trials with CDK4/6 inhibitors in patients with HR+, HER2− breast cancer in the adjuvant setting.

Clinical Trial	Patient Selection	Estimated Enrollment (*n*)	Therapeutic Regimen	Study Start Year	Estimated Completion Year	Clinical Trial Information
NATALEE	Pre- and postmenopausal women and men	5000	Ribociclib + endocrine therapy versus endocrine therapy alone	2018	2026	NCT03701334
APPALACHES	Women or men 70 years and older for whom chemotherapy is indicated	366	Palbociclib + endocrine therapy versus chemotherapy + endocrine therapy	2019	2025	NCT03609047
POLAR	Pre- and postmenopausal women or men with local/regional recurrence of BC	400	Palbociclib + endocrine therapy versus endocrine therapy alone	2019	2024	NCT03820830
POETIC-A	Postmenopausal woman with high 5-year risk of relapse	2500	Abemaciclib + endocrine therapy versus endocrine therapy alone	2020	2026	NCT04584853
LEADER	Pre- and postmenopausal women	120	Ribociclib + endocrine therapy versus endocrine therapy alone	2017	2022	NCT03285412
SAFIA	Pre- and postmenopausal women in the neoadjuvant setting	400	Palbociclib + fulvestrant versus placebo + fulvestrant alone	2018	2023	NCT03447132

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
