# Peer review of "Targeted Therapy in HR+ HER2− Metastatic Breast Cancer: Current Clinical Trials and Their Implications for CDK4/6 Inhibitor Therapy and beyond Treatment Options"

_cancers, 2021, doi:10.3390/cancers13235994_

Round 1

Reviewer 1 Report

I recommend this manuscript for publication of Cancers. 

Author Response

We are thankful for the positive review and your recommendation for publication in Cancers.

Reviewer 2 Report

In my original review, I proposed to include a figure describing the link between CDK4/6 activity Rb, Estrogen, PI3K/AKT/mTOR in light of the described text. The authors including a sentence on lines 61’ Excellent detailed schematic illustrations about these complex pathways have been shown in several earlier publications [7,8,9]’. I still believe that such a scheme can improve the paper.

An additional suggestion is to include bullet points of major conclusions, as it is very long and describe too many details and controversial aspects.

Author Response

We thank the reviewer for the encouraging answer. As suggested, we added a figure describing the link between CDK4/6 activity Rb, Estrogen, PI3K/AKT/mTOR in light of the described text.

Furthermore, we added bullet points in line 262, 317, and 343.

This manuscript is a resubmission of an earlier submission. The following is a list of the peer review reports and author responses from that submission.

Round 1

Reviewer 1 Report

The aim of the review paper entitled “ Targeted Therapy in HR+ HER2- Metastatic Breast Cancer: The Current Role of CDK4/6 Inhibitors and beyond Treatment Options” is to  summarize current knowledge associated with CDK4/6 targeting as therapeutic strategy for hormone receptors (HR)- and/or HER2-positive metastatic breast cancer. The review includes 5 tables describing ongoing clinical trials targeting CDK4/6 either alone or in combination as well as selected trials targeting PI3K. In general, the review is well written and could be informative.

Yet there are several critical points to address and improve the current version.

  • There 5 different tables that might be combined and may especially 2&3, or even all except 4.
  • The rational of selecting patients that could benefit of CDK4/6 inhibition is not explained well (line 171)

In addition, may consider to change the subtitle “ who benefits the most?”

  • Similarly, subtitle 2: “The rationale of indicating a CDK4/6 inhibitor therapy” May consider:

“The rationale of targeting CDK4/6 for breast cancer therapy”.

It will be important to include explanation on the rationale of using CDK4/6 inhibition.

Line 57 “ estrogen acts as a stimulator to this cascade” (of this cascade)- explain how.

It is known that in HR+ breast cancer, cyclin D is highly expressed and loss of pRb is rare, making the G1-to-S checkpoint an ideal therapeutic target, this information is not included.

What is the rationale of HER2+?

  • A figure describing the link between CDK4/6 activity Rb, Estrogen, PI3K/AKT/mTOR in light of the described text, could markedly improve the paper.
  • Subtitle 3- Consider to change to “Clinical impact of CDK4/6 targeting”
  • Subtitle 8- Consider to omit the “old”

Line 32: I would change “a central element” to “a major approach/strategy”.  

Line 64: I would delete the text within the brackets

Line 66: “Abemaciclib is also approved as an monotherapeutic agent” please specify to which patients.

Line 104: Which subgroup of TNBC patients could be benefit of this treatment?

Line 155/6: It is unclear “have to keep in mind the selection bias and  responder bias” please explain.

Reviewer 2 Report

Major issues: 

The manuscript is well written and provides a good review of current studies. However, a section on the significance of ESR1 mutations, particularly for therapy selection, is missing.

Minor issues:

Line 21: Summary instead of summery

Line 65: significance of ESR1 mutations is missing

Line 91: HER2-directed instead of Her2-directed

Table 3: "versus" instead of "verus"

Line 152: CDK4/6 inhibitors instead of CK4/6 inhibitors

Line 173: HER2- instead of Her2-

Line 179: "Progression of disease under endocrine therapy" instead of “endocrine-resistant tumors”

Line 215: “almost all” instead of “all”

Line 253: PFS instead of PSF

Table 4: HER2 instead of Her2

Line 341: HER2 instead of Her2

Reviewer 3 Report

The authors of this manuscript aim to summarize and give an updated information about the clinical relevance, patient selection, and ongoing trials of CDK4/6 inhibitors and further targeted therapy options. The authors state that the manuscript focuses on clinical aspects and practicability as well as adverse events from the clinical trials. The manuscript carries good among of information about current clinical trials and its relevance with the developing inhibitors. However, the manuscript overall lacks fluent flow of the logic and is broad, as it is difficult to understand what the authors are trying to state throughout.

  1. The subheadings be improved to contain the message of the paragraph clearer for the readers to understand the flow of the review.
  2. The authors must improve the overall flow of the manuscript, especially between the subheadings. Perhaps changing the orders of the paragraphs or removing them could help improve the manuscript. Some of the subheadings (such as 6, 7, and 8) could be made shorter or combined into one to improve the flow and keep the focus on CDK4/6 inhibitors throughout the manuscript.
  3. The conclusion of the manuscript should be improved.
  4. The authors must do an overall review with the typos and proofread the overall English manuscript. (Abstract has typos as well.)
  5. The title of the manuscript should be changed to suggest that the manuscript carries information about clinical trials and its implications. Current title may be misleading.